# The Role of Nutritional Therapy in the Treatment of Adults with Crohn’s Disease: A Review

**DOI:** 10.3390/nu17203186

**Published:** 2025-10-10

**Authors:** Raffaele Li Voti, Fabio Salvatore Macaluso, Elena Banci, Angelo Campanozzi, Giulia D’Arcangelo, Alessia De Blasi, Salvatore Oliva, Elena Sofia Pieri, Sara Renzo, Cosimo Ruggiero, Giusy Russo, Luca Scarallo, Paolo Lionetti, Ambrogio Orlando

**Affiliations:** 1Inflammatory Bowel Disease Unit, “Villa Sofia-Cervello” Hospital, 90146 Palermo, Italy; raffo.livoti@gmail.com (R.L.V.); ambrogiorlando@gmail.com (A.O.); 2Dietetics Unit, Meyer Children’s Hospital IRCCS, 50139 Florence, Italy; elena.banci@meyer.it; 3Pediatrics, Department of Medical and Surgical Sciences, University of Foggia, 71100 Foggia, Italy; angelo.campanozzi@unifg.it (A.C.); giuliadarcangelo87@gmail.com (G.D.); elenasofia.pieri@gmail.com (E.S.P.); 4Gastroenterology and Nutrition Unit, Meyer Children’s Hospital IRCCS, 50139 Florence, Italy; alessia.deblasi@meyer.it (A.D.B.); sara.renzo@meyer.it (S.R.); luca.scarallo@gmail.com (L.S.); paolo.lionetti@unifi.it (P.L.); 5Pediatric Gastroenterology and Liver Unit, Maternal and Child Health Department, Sapienza University of Rome, 00185 Rome, Italy; salvatore.oliva@uniroma1.it (S.O.); cosimo.ruggiero@uniroma1.it (C.R.); giusy.russo@uniroma1.it (G.R.); 6Department of Neurofarba, University of Florence, 50139 Florence, Italy

**Keywords:** adults, Crohn’s disease, Crohn’s disease exclusion diet, Mediterranean diet, nutritional therapy

## Abstract

Crohn’s disease (CD) is an immune-mediated inflammatory bowel disease (IBD) with a multifactorial pathogenesis involving genetic predisposition, immune dysregulation, and environmental triggers. Dietary patterns have recently garnered growing attention for their potential benefits and risks in patients with IBD. Nutritional therapy has been established as an effective option in pediatric populations, but its role in adults remains less defined. The available studies indicate that while no single diet can be universally recommended, adherence to a Mediterranean diet is associated with multiple health benefits. Nutritional therapy appears promising in inducing clinical remission in adults with mild to moderate CD, particularly when partial enteral nutrition is combined with food-based modifications. Tailoring these strategies to cultural contexts and providing support from qualified dietitians may improve adherence, clinical outcomes, and overall quality of life. This review highlights the growing role of nutritional therapy in adult CD and its potential integration into routine management alongside conventional treatments.

## 1. Introduction

Crohn’s disease (CD) is an immune-mediated condition that can lead to progressive bowel damage and long-term disability [1]. It develops in genetically predisposed individuals in response to lifelong environmental exposures, collectively referred to as the exposome [2]. Among these, diet has gained particular attention given its ability to modulate the intestinal microbiota and immune regulation. The increasing incidence of CD in Western countries has raised concerns about the impact of a Westernized diet, characterized by the high consumption of certain food types and additives [3]. This trend is mirrored in the rising prevalence of IBD in rapidly developing countries adopting Western dietary habits, as well as among immigrants from low-incidence regions who relocate to Western countries [4].

Although no specific diet is currently recommended for patients with established IBD [5], substantial pre-clinical and epidemiological evidence indicates that nutrition significantly influences both disease onset and progression [6,7]. Alterations in the gut microbiota of patients with CD are well documented. Several studies indicate that dysbiosis may be a risk factor for disease onset rather than merely a consequence of intestinal inflammation [8]. Supporting this hypothesis, altered gut microbiota profiles have been identified in unaffected relatives of CD patients [9]. Dysbiosis has also been correlated with elevated inflammatory markers in patients with CD [10].

Dietary components typical of the Western diet have been associated with impaired intestinal barrier function, disrupted microbiome composition and function, and consequent immune dysregulation [6,7,11]. In particular, low fiber intake and high consumption of simple carbohydrates and polyunsaturated fatty acids (PUFAs) are linked to a reduction in butyrate-producing bacteria and decreased microbial diversity [12]. Food additives—including sweeteners, artificial colors, emulsifiers, and stabilizers—have been shown in animal models to promote gut inflammation, induce dysbiosis, and increase intestinal permeability [6,13,14]. Conversely, high-fiber diets appear to be associated with a reduced risk of developing CD [15]. Collectively, these dietary factors may contribute substantially to IBD susceptibility and to disease flares or progression, as demonstrated in diverse populations worldwide [16,17].

Dietary considerations remain relevant after diagnosis. Many patients modify their diet independently, often without adequate knowledge or professional guidance, leading to unnecessary food restrictions and micronutrient deficiencies [18,19,20,21]. In this context, interest in diet and nutrition has grown considerably in recent years, with studies exploring nutrition-based interventions aimed at improving disease control and enhancing the effectiveness of conventional therapies. While no single dietary approach has yet achieved sufficient evidence to be incorporated into current clinical guidelines [5], certain nutritional strategies—particularly in children and young adults—have shown promising efficacy in the treatment of CD. More recently, some studies have also assessed the potential of nutritional therapy in adult patients, with encouraging initial evidence.

We conducted a comprehensive PubMed and clinical trials registries literature search up to May 2025 to evaluate current evidence on nutritional interventions in adults with CD. The following keywords were used in the literature search: (Adults [Mesh] OR adult patients) AND (Inflammatory bowel disease [Mesh] OR Crohn’s disease) AND exclusive enteral nutrition OR Crohn’s disease exclusion diet OR specific carbohydrate diet OR FODMAP OR Mediterranean diet OR Low Emulsifier Diet OR nutritional therapy. Articles not published in the English language were excluded. The bibliographic online search was complemented by a manual review of the relevant references.

This narrative review aims to synthesize current evidence on the use of nutritional therapy in adults with CD, to identify key challenges in its implementation, and to explore potential strategies to address these barriers (Figure 1).

## 2. Exclusive Enteral Nutrition

Exclusive enteral nutrition (EEN) was the first nutritional therapy proposed and supported by robust evidence for its efficacy in inducing remission in patients with CD, with initial reports dating back to the 1980s [22]. EEN is now a well-established first-line therapy for mild to moderate pediatric CD, as recommended by current guidelines [23]. It consists of the exclusive intake of a complete liquid nutritional formula over approximately 6–8 weeks, achieving high rates of both clinical and endoscopic remission. Formulas vary in composition and can be classified as elemental, semi-elemental, or polymeric [24]. Multiple studies comparing amino acid concentrations or formula compositions have found minimal or no differences in efficacy [25,26]. Consequently, no single composition is universally recommended, although international guidelines favor polymeric formulas due to their better tolerability [24,27].

The exact mechanism of action of EEN remains unclear. Proposed mechanisms include modulation of the inflammatory milieu through reduced production of pro-inflammatory cytokines such as IL-6 and IL-8, regulation of the NF-κB pathway, and enhancement of intestinal barrier integrity [28,29].

Given the strong pediatric evidence base, recent studies have investigated the use of EEN in adults, either as monotherapy or in combination with corticosteroids, immunosuppressants, or biologic agents. Wall et al. demonstrated clinical benefit in a prospective pilot study [30], while Sharma et al. reported similar findings in a retrospective cohort following an 8-week course of EEN—although nearly 20% of patients were unable to tolerate the regimen [31]. Better adherence rates were observed by Shukla et al., who conducted a prospective study with the support of a qualified dietitian, achieving good clinical remission rates, particularly in colonic or ileocolic disease, and demonstrating the value of professional dietary guidance [32]. Comparable outcomes were reported by Ramaswamy et al., with clinical remission rates of 61% and biochemical remission rates—defined as the normalization of fecal calprotectin and C-reactive protein (CRP)—of 56%, with higher success rates in patients who completed the EEN course; no significant difference was observed with concomitant corticosteroid use [33]. Despite these promising results, it is important to note that the study included only 66 patients, with considerable heterogeneity in disease phenotype and concurrent medications. Participants were undergoing different treatments, including biologics and immunosuppressants, and over 40% were on corticosteroids during the study period.

A recent open-label prospective study from China reported a clinical response rate of approximately 73% in patients with active CD who completed a 12-week course of EEN without concomitant medication [34]. This study also demonstrated endoscopic response in 50% of patients and transmural healing in around 16%. Flow cytometry analysis revealed an early reduction in activated neutrophils and monocytes, inhibition of the Th17 pathway, and modulation of B-cell immune responses. Notwithstanding the important findings, the study had a small sample size, including only twenty patients, and required nasogastric tube placement—an intervention that may limit adherence in larger patient populations.

EEN has also been evaluated in specific clinical settings—such as during pregnancy, in patients with abdominal fistulas, and in preoperative contexts—showing excellent safety and encouraging results [35,36,37].

Despite these promising outcomes, limitations of the current evidence include small sample sizes, heterogeneous inclusion criteria, variations in formula composition, frequent absence of endoscopic assessment, and lack of control groups. Notably, a meta-analysis by Narula et al. found comparable efficacy between EEN and corticosteroids for the induction of remission in pediatric CD, but lower efficacy in adults [38]. Furthermore, a Japanese study failed to demonstrate superiority of combination therapy with EEN and anti-TNF agents over anti-TNF monotherapy in maintaining remission in patients who had responded to induction with biologics [39]. These mixed findings likely explain why EEN has not yet been incorporated into international guidelines for the treatment of adult CD and indicate that long-term, larger, and preferably randomized studies are needed to further validate this nutritional therapy in adults with CD.

## 3. Partial Enteral Nutrition and Exclusion Diets

To address the limited adherence typically observed with EEN, partial enteral nutrition (PEN) has been proposed as an alternative. PEN provides only a proportion of daily caloric requirements through an enteral formula, with the remainder supplied by regular food intake [40]. Although more tolerable and easier to follow than EEN, early studies pairing PEN with an unrestricted diet failed to demonstrate comparable efficacy [41,42], and PEN was therefore not recommended alongside EEN in pediatric or adult guidelines for CD.

More recently, combining PEN with structured exclusion diets—most notably the Crohn’s Disease Exclusion Diet (CDED) and the Crohn’s Disease Treatment-with-Eating Diet (CD-TREAT) [43,44]—has shown efficacy approaching that of EEN [40,41]. Initially studied in children, PEN with exclusion diets has subsequently been applied in adult cohorts with encouraging results, although adherence rates tend to be lower than in pediatric populations [40,45].

CDED is a whole-food diet incorporating mandatory items and progressing through three increasingly liberal phases, designed to reduce exposure to potentially pro-inflammatory dietary components that may influence gut microbiota composition, immune activation, and epithelial barrier integrity [46]. In the pilot study by Sigall-Boneh et al. [43], 13 young adults (mean age 19.3 ± 3.9 years) with prior medical therapy failure received CDED plus PEN using Modulen^®^ (Nestlé, Vevey, Switzerland), a nutritionally complete polymeric formula enriched with transforming growth factor-β2, for 6 weeks. Clinical remission was achieved in 69.2% of patients. These findings were replicated in 2017 in 21 patients (11 adults) with secondary loss of response to biologic therapy, where 61.9% achieved clinical remission at week 6, accompanied by reductions in Harvey–Bradshaw Index (HBI) from 9.4 ± 4.2 to 2.6 ± 3.8 (*p* < 0.001) and C-reactive protein (CRP) from 2.8 ± 3.4 to 0.7 ± 0.5 mg/dL (*p* = 0.005) [47]. Similarly, Szczubelek et al. [48] reported clinical remission in 76.7% of 32 adult patients after 6 weeks of CDED plus PEN, rising to 82.1% at week 12, with significant reductions in fecal calprotectin; adherence was high, with only 4 of 32 patients failing to follow the protocol.

In 2022, Yanai et al. [45] conducted the first randomized, open-label trial comparing CDED alone versus CDED plus PEN in biologic-naïve adults with mild-to-moderate CD. Forty-four patients were randomized, with week 6 remission rates of 66.7% and 73.7%, respectively. At week 24, sustained remission favored the CDED plus PEN group (63.2% vs. 38.2%), although this difference did not reach statistical significance. Both groups showed similar CRP and fecal calprotectin improvements at week 12, but endoscopic remission was more frequent with CDED plus PEN (42.1% vs. 28.6%). Although underpowered, the trial suggests potential superiority of the combined approach.

Other studies have provided additional insights. Pasta et al. [49] found that CDED (with PEN only in malnourished patients) resulted in significantly lower median HBI than a Mediterranean diet after 24 weeks (2 [IQR 1–3] vs. 5 [IQR 4–7], *p* < 0.0001), although no differences in CRP or fecal calprotectin were observed. In newly diagnosed, treatment-naïve patients, Ovadia et al. [50] compared PEN plus a standard diet with budesonide over 8 weeks, observing greater improvement in CDAI and Lewis score in the PEN group, while CRP and fecal calprotectin tended to improve more with budesonide.

The CD-TREAT diet, designed to replicate the nutrient profile of EEN while excluding specific components such as gluten, lactose, and alcohol [44], has demonstrated clinical efficacy and high adherence in children but remains underexplored in adults. Similarly, the McMaster Elimination Diet for CD (MED-CD) was evaluated by Narula et al. [51] after a 2-week EEN induction in 13 adult patients; at week 12, 38.5% maintained clinical remission, 46.2% achieved endoscopic response, and 15.4% reached endoscopic remission, with adherence of 67%.

Although PEN appears to be a more tolerable and potentially effective alternative to EEN, the current evidence has several key limitations—similar to those of EEN—including the small sample size, short follow-up period, absence of large randomized controlled trials, limited endoscopic data, and lack of control groups incorporating standard medical therapy.

Moreover, despite being less restrictive than EEN, both PEN and exclusion diets face adherence challenges. Strategies to improve long-term compliance include virtual support tools, app-based monitoring, and close involvement of specialist dietitians, as demonstrated by Dutch and Australian cohort studies [52,53]. Boneh et al. [54] further showed that adapting CDED to local food cultures and social traditions can enhance acceptability. A multicenter randomized trial in Italy is currently comparing standard CDED with a Mediterranean-adapted version (MED-CDED), aiming to improve adherence while preserving efficacy. The MD, already favored by the American Gastroenterological Association for its systemic health benefits, may be an optimal base for such adaptation [5].

In summary, PEN combined with structured exclusion diets—particularly CDED—represents a promising dietary strategy for induction of remission in mild-to-moderate CD. While efficacy appears comparable to EEN in selected contexts, adherence remains a limitation. Integration of dietitian-led guidance, cultural adaptation, and patient-centered support tools may enhance long-term feasibility. Further adequately powered randomized trials, with a focus on endoscopic outcomes and extended follow-up duration, are warranted to define the role of PEN plus exclusion diets as both standalone and adjunctive therapy in adult CD.

## 4. Specific Carbohydrate Diet

The Specific Carbohydrate Diet (SCD) is a nutritional approach first proposed by Dr. Sidney Haas in 1951 for the treatment of coeliac disease [55]. In 1994, biochemist Elaine Gottschall advocated for its use in treating ulcerative colitis (UC), inspired by her personal experience with treating her daughter [56].

The rationale for the SCD is based on the hypothesis that bacterial overgrowth and excessive fermentation of undigested carbohydrates in IBD contribute to small bowel mucosal injury. The resulting organic acids may exacerbate inflammation, perpetuating a cycle of mucosal damage [56,57]. SCD is a restrictive nutritional approach that eliminates disaccharides and most polysaccharides to reduce carbohydrate malabsorption, limit fermentation, and mitigate pro-inflammatory effects [56].

SCD has been predominantly studied in pediatric populations, where it has shown positive effects on symptoms and inflammatory markers in patients with CD [58,59,60]. However, evidence for endoscopic improvement remains inconsistent. Wabbeh et al. did not observe mucosal healing in SCD-treated patients, in contrast to earlier findings by Suskind et al. [61,62]. Moreover, the multicenter PRODUCT study compared SCD, a modified version of SCD, and a standard diet, finding no significant differences between groups in primary outcomes. Nevertheless, some patients in the SCD and modified SCD groups reported symptomatic improvement and reductions in fecal calprotectin [63].

In adult patients with CD, data remain limited. An online survey of 417 adults (median age 34.9 ± 16.4 years; approximately half with CD) found self-reported symptomatic remission with SCD in 33% at 2 months and in 42% at both 6 and 12 months, though the duration of remission varied from 2 weeks to over 3 months [64]. The DINE-CD trial, a multicenter randomized study conducted in the United States, compared SCD with the MD in adults with CD, showing no statistically significant differences between the two interventions [65]. Adherence rates and microbiome diversity were similar in both groups. A Massachusetts group has proposed a modified and augmented version of SCD for adults with IBD, reporting potential benefits in achieving symptomatic remission and reducing the need for standard pharmacologic therapy [66]. However, this approach also underscored the challenges of sustaining adherence to SCD or derivative dietary regimens.

As a highly restrictive diet, SCD presents significant adherence barriers, compounded by the lack of clear guidance from healthcare providers [67]. Given these limitations and the absence of consistent evidence for mucosal healing, SCD has not been widely adopted as a therapeutic approach for CD and is not currently recommended in international clinical guidelines.

## 5. Mediterranean Diet

The term Mediterranean diet (MD) was coined by Ancel Keys in 1960 to describe the dietary habits of populations living along the Mediterranean coast [68]. Although subsequent studies showed that this pattern is more characteristic of certain Greek islands and Southern Italy than of other Mediterranean regions, MD remains one of the most extensively studied and recognized dietary models worldwide [69].

MD is characterized by a high consumption of unrefined cereals, fresh fruits and vegetables, nuts, legumes, potatoes, and low-fat dairy products; an abundant use of olive oil; and a moderate intake of fish, poultry, and red meat [69,70]. Since the 1960s, adherence to MD has been associated with significant benefits in patients with cardiovascular disease, type 2 diabetes, metabolic syndrome, and certain cancers [69,71,72]. These effects are likely related to its anti-inflammatory properties, which are not yet fully elucidated but appear to involve downregulation of several pro-inflammatory cytokines, including IL-1β, IL-6, and TNF-α [70,73].

Given its anti-inflammatory potential and systemic health benefits, MD represents an attractive dietary option for patients with IBD, particularly CD. However, only a limited number of studies have specifically evaluated MD in adults with CD. Chicco et al. reported that six months of MD improved nutritional status and hepatic steatosis in CD patients, with additional improvements in CDAI scores, CRP, and fecal calprotectin [74]. As noted earlier, Lewis et al. found comparable efficacy between MD and the Specific Carbohydrate Diet (SCD) in adults with CD [65], further supporting its potential value. In this randomized controlled trial, the primary endpoint was clinical remission, with secondary endpoints including improvements in fecal calprotectin and CRP. Remission occurred in 43.5% of patients on MD and 46.5% on SCD, while improvements in fecal calprotectin were seen in 30.8% and 34.8%, respectively.

Most recently, Godny et al. conducted a prospective longitudinal study involving over two hundred adult patients with newly diagnosed CD [75]. Patients received regular follow-up by a dietitian, and those with better adherence to MD exhibited higher rates of non-complicated disease, along with lower levels of CRP, fecal calprotectin, and microbial dysbiosis markers.

Due to its well-documented health benefits, relatively high adherence rates (especially in certain geographical areas), and preliminary evidence of clinical benefit in IBD, MD is currently considered by many experts as the preferred dietary regimen for patients with CD [5,76]. Moreover, combining MD with other effective dietary strategies, such as the CDED, offers an appealing opportunity to enhance adherence while potentially synergizing the positive effects of both approaches.

Despite these undeniable benefits, further research is warranted to confirm the long-term efficacy and tolerability of this approach in patients with CD, both in combination with PEN or standard medical therapy, and in those with longer disease duration. It would also be important to assess the feasibility of this nutritional therapy in countries where the MD is not part of the standard dietary pattern, similar to what has been proposed for the CDED.

## 6. Low Fermentable Oligosaccharides, Disaccharides, Monosaccharides, and Polyols Diet

Fermentable oligosaccharides, disaccharides, monosaccharides, and polyols (FODMAPs) are short-chain carbohydrates that increase small intestinal water content through osmotic effects (e.g., fructose, mannitol) and promote colonic gas production via microbial fermentation (e.g., fructans, galacto-oligosaccharides) [77]. FODMAPs are known to exacerbate symptoms in patients with irritable bowel syndrome (IBS) [78] and may trigger similar symptoms in patients with CD, particularly those with overlapping IBS.

Several studies have shown that short-term adherence to a low FODMAP diet can improve symptoms and quality of life in patients with quiescent or mild-to-moderate CD who also meet Rome criteria for IBS [79,80,81]. In 2020, Cox et al. conducted a randomized trial in patients with UC or CD in clinical remission but experiencing IBS-like symptoms [76]. A 4-week low FODMAP intervention improved symptom control compared with a control diet, although no differences were observed in inflammatory markers.

Similarly, Bodini et al. performed a randomized trial comparing a 6-week low FODMAP diet with a standard diet in patients with quiescent or mild disease meeting Rome IV criteria [82]. At the end of the intervention, median HBI scores decreased in the low FODMAP group (from 4; IQR, 3–5 to 3; IQR, 2–3; *p* = 0.024), but not in the standard diet group (from 3; IQR, 3–3 to 3; IQR, 2–4). Interestingly, a small but statistically significant reduction in fecal calprotectin was also observed in the low FODMAP group (from 76.6 mg/kg; IQR, 50–286.3 to 50 mg/kg; IQR, 50.6–81; *p* = 0.004), with no corresponding change in the standard diet group. These findings are consistent with those of a small randomized controlled trial conducted by Pedersen et al. in 2017 [83].

In summary, a low FODMAP diet appears effective in alleviating symptoms and improving quality of life in adults with quiescent or mild CD and overlapping IBS, even though longer-term studies are needed to confirm this benefit, but it exerts minimal or no impact on inflammatory markers or overall disease activity [84].

## 7. IBD-MAID and Low-Emulsifier Diet (LED)

Food additives—such as non-nutritive sweeteners, nitrites/nitrates from processed or smoked meats, maltodextrin, and emulsifiers—including polysorbate 80, carboxymethylcellulose, and carrageenan gum—have been implicated in promoting gut inflammation and dysbiosis [6,13,14]. A growing body of preclinical and translational evidence supports this association, particularly regarding their impact on microbial diversity. The ENIGMA study demonstrated that polysorbate 80 reduces the growth of *Faecalibacterium prausnitzii*, a beneficial commensal strain, while promoting the expansion of *Proteobacteria* [85]. More recently, Rosta et al. reported that carboxymethylcellulose exerts an even more detrimental effect on microbial diversity in murine models [86]. Similarly, carrageenan has been shown to increase the abundance of pro-inflammatory microbial strains, such as *Bacteroides* and *Pseudomonas,* both in animal and human models [14].

Supported by these findings, Bancil et al. conducted a randomized trial—the ADDapt study—designed to evaluate the effects of a low-emulsifier diet (LED) compared to a control diet over eight weeks in 154 patients with mild to moderately active CD [87]. After eight weeks, patients following the LED were more than twice as likely to experience clinical remission (Adjusted RR 2.1; 95% CI 1.0–4.4; *p* = 0.042) and a >50% reduction in fecal calprotectin levels (Adjusted RR 2.9; 95% CI 1.1–8.0; *p* = 0.039). A smaller randomized trial by Fitzpatrick et al. compared the LED with a high-emulsifier diet in 24 adult patients with CD, all of whom were receiving stable medical therapy [88]. After four weeks of nutritional intervention, no significant differences were observed between the two groups in terms of clinical or radiological improvement, as assessed by intestinal ultrasound. Despite the conflicting outcomes, the disparity in sample size and follow-up between these two trials should be carefully considered when interpreting these findings.

An interesting recent nutritional intervention proposed by an Australian research group compared a healthy dietary pattern with a targeted regimen aimed at reducing the intake of food additives (IBD-MAID) [89]. This pilot study demonstrated good adherence to the dietary protocol, with improvement in symptoms—particularly among patients with CD—as well as favorable changes in inflammatory markers. Given these preliminary findings, larger-scale studies with prolonged follow-up durations are warranted to better assess the efficacy of this dietary approach in patients with CD.

## 8. Conclusions and Future Directions

Nutrition plays a pivotal role in the life and disease management of adult patients with CD, with a substantial impact on quality of life. Properly addressing nutritional needs and implementing targeted nutritional therapies can enhance quality of life, improve nutritional status, and support clinicians in achieving clinical remission, normalization of fecal calprotectin and CRP levels, and potentially endoscopic remission (Table 1).

Although not consistently included in all international guidelines, available evidence indicates that nutritional therapy may induce a therapeutic response in patients with mild to moderate CD, either as monotherapy or in combination with conventional treatments, and may also enhance outcomes in more severe cases, as already demonstrated in pediatric populations.

Thus, nutritional therapy may represent a safe and valuable approach to enhance and support treat-to-target strategies, both by improving the efficacy of standard medical therapy and by preventing unnecessary escalation. In addition, it may help achieve not only the short- and intermediate-term goals outlined in the STRIDE II consensus [90] but also its ultimate targets: normalization of quality of life and absence of disability.

Current evidence suggests that certain nutritional interventions in adults with CD may be both more effective and better tolerated. Although only a limited number of studies have directly compared different strategies, the MD and PEN—with or without exclusion diets, particularly CDED—appear to provide the most favorable balance between patient adherence and clinical efficacy, particularly in Western populations. Moreover, PEN has shown potential for endoscopic improvement, though the available data remain limited. Other dietary interventions have demonstrated only clinical benefits and often rely on strict regimens that are difficult to maintain over the long term or may reinforce avoidance behaviors, as observed with the SCD and low-FODMAP approaches. Similarly, although some strategies have shown promising effects on clinical outcomes, endoscopic response, and inflammatory markers, they may still pose challenges for long-term adherence and fail to support normalization of quality of life, as in the case of EEN. Newer approaches, such as LED and IBD-MAID, while conceptually compelling and supported by encouraging preclinical data, remain too recent and currently lack sufficient evidence to justify their use over more extensively studied interventions such as PEN or MD.

Finally, many of the studies conducted to date suffer from small sample sizes and/or lack of control groups, and endoscopic assessment has rarely been performed. Given the clinical relevance of this approach in CD and its therapeutic potential, further research is warranted to better define the optimal indications for nutritional therapy and the most effective strategies to improve both therapeutic response and adherence. This should include a comprehensive evaluation of all treatment goals, as recommended by the STRIDE II consensus [90].

As a self-administered, daily intervention, nutritional therapy offers the opportunity to provide continuous benefits in terms of patient well-being and disease control. Beyond modulating the immune response, it can increase patient awareness of the disease and foster greater engagement in clinical decision-making and treatment adherence. Support from expert dietitians, complemented by technological tools, is essential to guide patients throughout the therapeutic process and optimize adherence. Adapting these dietary approaches to different cultural and culinary traditions is equally important to facilitate their adoption across diverse CD populations and achieve dissemination comparable to that of conventional and biologic therapies.

In conclusion, nutritional therapy in CD has the potential to further advance disease control, contribute to overcoming the current therapeutic ceiling, and promote a more holistic approach to disease management.

## Figures and Tables

**Figure 1 nutrients-17-03186-f001:**
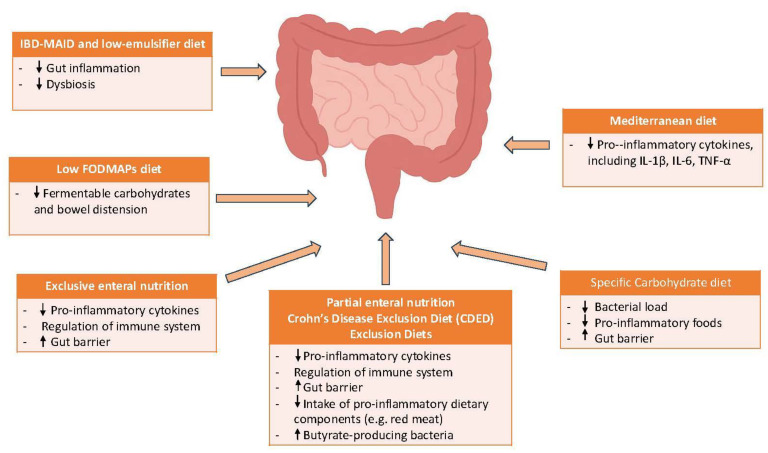
Pathophysiological mechanisms linking dietary interventions to clinical outcomes in Crohn’s disease.

**Table 1 nutrients-17-03186-t001:** Comparison of nutritional approaches in adult patients with Crohn‘s disease.

Nutritional Therapy	Rationale	Pros	Cons	RCTs	CR *	BR	ER *	References
**EEN**	Reduction in pro-inflammatory cytokines, regulation of the immune system, and enhancement of the gut barrier	Robust evidence (especially in children), good safety also in specific settings (pregnancy, fistulas)	Low adherence, apparently not superior to corticosteroids	✔	✔	✔	✔	Wall CL [30]; Sharma S [31]; Shukla D [32]; Kakkadasam Ramaswamy P [33]; Diao N [34]; Escuro AA [35]; Yang Q [36]; Wall CL [37]
**PEN/CDED/ED**	Same as EEN plus reduction in intake of pro-inflammatory dietary components (e.g., red meat), reducing gut microbiota, increasing butyrate-producing bacteria	More tolerated than EEN, data from multiple RCTs, possibility of adapting dietary regimen to different cultures and nutritional habits	Need for a physician and a dietitian guide to maintain a correct diet; few data on endoscopic response or maintenance of remission	✔	✔	✔	✔	Sigall Boneh R [43], Svolos V [44], Yanai H [45]; Sigall Boneh [47]; Szczubełek M [48]; Pasta A [49]; Ovadia B [50]; Narula N [51]
**SCD**	Reduction in bacterial load and pro-inflammatory diet that may lead to bowel injuries and a reduction in gut barrier integrity	Swift symptomatic response	Strict diet with risk of nutritional deficiencies if not followed up by expert dieticians; not assessed the benefits on endoscopic response	✔	✔	✔	✘	Suskind DL [64]; Lewis JD [65]; Olendzki BC [66]
**MD**	Downregulation of several pro-inflammatory cytokines, including, among others, IL-1β, IL-6, and TNF-α	Easier adherence, low risk of nutritional deficiencies, known benefits also in the cardiovascular and metabolic areas	Not assessed the benefits of the endoscopic response	✔	✔	✔	**NA**	Lewis JD [65]; Chicco F [74]; Godny L [75]
**Low-FODMAPs**	Reduction in fermentable carbohydrates in small intestinal water through gas production and colonic distension through microbial fermentation	Swift symptomatic response, even in patients with overlapping IBS	No benefits on biochemical, and no benefits on endoscopic response	✔	✔	✘	**NA**	Cox SR [77]; Prince AC [80]; Bodini G [82]; Pedersen N [83]
**IBD-MAID and LED**	Reduction in gut inflammation and dysbiosis by eliminating food additives	Strong rationale, possible systemic benefits in other health-related areas	Very little data on efficacy, possible difficulty in managing a diet with low food additives	✔	✔	✔	**NA**	Bancil A [87]; Fitzpatrick JA [88]; Marsh A [89]

* Clinical response and endoscopic response definitions vary between studies, with high heterogeneity. RCTs: Randomized Controlled Trials; CR: Clinical response; BR: Biochemical response; ER: Endoscopic response; EEN: exclusive enteral nutrition; PEN: Partial enteral nutrition; CDED: Crohn’s Disease Exclusion Diet; ED: Exclusion diets; SCD: Specific Carbohydrate diet; MD: Mediterranean Diet; Low-FODMAPs: Low Fermentable Oligosaccharides, Disaccharides, Monosaccharides and Polyols; LED: Low Emulsifier diet; NA: not assessed.

## Data Availability

No new data were created for this manuscript.

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
