# Peer review of "The Role of Nutritional Therapy in the Treatment of Adults with Crohn’s Disease: A Review"

_nutrients, 2025, doi:10.3390/nu17203186_

Round 1
Reviewer 1 Report
Comments and Suggestions for Authors
This review provides a comprehensive and honest assessment of nutritional therapy in adult Crohn's disease, serving as a valuable reference point for both clinicians and researchers. The authors appropriately acknowledge limitations while offering specific directions for future research.
However, as a review manuscript rather than experimental research, the acknowledged limitations in sample sizes and follow-up periods of the included studies necessitate further validation. Given these methodological constraints explicitly noted by the authors themselves, I believe this manuscript requires additional evidence before it can be considered for publication in its current form
Author Response
Reviewer 1: This review provides a comprehensive and honest assessment of nutritional therapy in adult Crohn's disease, serving as a valuable reference point for both clinicians and researchers. The authors appropriately acknowledge limitations while offering specific directions for future research. However, as a review manuscript rather than experimental research, the acknowledged limitations in sample sizes and follow-up periods of the included studies necessitate further validation. Given these methodological constraints explicitly noted by the authors themselves, I believe this manuscript requires additional evidence before it can be considered for publication in its current form
Response: We sincerely thank the Reviewer for the thoughtful and constructive feedback on our manuscript. With regard to the need for further validation and additional evidence, we would like to clarify that our review includes all currently available and relevant evidence on nutritional therapy in adult Crohn’s disease, based on a comprehensive literature search. To the best of our knowledge, there are no additional studies that meet our inclusion criteria or that have emerged since the completion of our review. We fully acknowledge the methodological limitations of the included studies—such as small sample sizes and short follow-up periods—which we have explicitly discussed in the manuscript. In response to the Reviewer’s comment, we have further strengthened the limitations of the current evidence throughout the manuscript and emphasized the need for high-quality, large-scale, and long-term studies to better define the role of nutritional interventions in adult Crohn’s disease.
Reviewer 2 Report
Comments and Suggestions for Authors
This review explores the evidence surrounding nutritional interventions for adults with Crohn’s disease, with particular emphasis on exclusive enteral nutrition, partial enteral nutrition combined with exclusion diets, the Mediterranean diet, and other elimination strategies. The authors synthesize findings from recent trials and highlight both the therapeutic potential and adherence challenges associated with these dietary approaches. They conclude that nutritional therapy, while less established in adults than in pediatrics, represents an increasingly important adjunct to conventional treatments.
The topic is timely and clinically relevant, as dietary management in adult Crohn’s disease remains underexplored compared to pediatric cohorts. The manuscript is overall well referenced and structured, with clear sections devoted to different nutritional strategies. That said, the tone of the review is more descriptive than analytical, and the critique of available evidence could be sharper.
Specific strengths: Comprehensive coverage of the main nutritional interventions currently studied in Crohn’s disease; integration of mechanistic insights (e.g., microbiome, barrier integrity) with clinical outcomes; attention to cultural and adherence factors, which are often neglected in reviews.
Major issues: The review frequently summarizes studies without adequately dissecting their methodological limitations (small sample sizes, lack of endoscopic endpoints, heterogeneity in populations). A more incisive appraisal would improve the manuscript’s value; while several diets are described in parallel, there is little effort to compare their relative strengths, limitations, and real-world applicability. A more direct head-to-head discussion would help clinicians; the review mentions IBD-MAID and the role of additives, but this could be expanded with a dedicated discussion on low emulsifier diets. There is growing preclinical and translational evidence that emulsifiers such as polysorbate 80 and carboxymethylcellulose may exacerbate gut inflammation and dysbiosis. Given the clinical momentum in this area, this omission feels important; the review cites STRIDE II, but the practical implications for integrating diet into treat-to-target strategies could be more explicitly stated.
Minor issues: The abstract and introduction could be tightened to reduce repetition; some sections, particularly on the Specific Carbohydrate Diet, are overly detailed given the limited evidence in adults, while other sections (Mediterranean diet, emulsifiers) are comparatively thin; table 1 is useful but could benefit from clearer definitions of CR/BR/ER for non-specialist readers.
Author Response
Reviewer 2: This review explores the evidence surrounding nutritional interventions for adults with Crohn’s disease, with particular emphasis on exclusive enteral nutrition, partial enteral nutrition combined with exclusion diets, the Mediterranean diet, and other elimination strategies. The authors synthesize findings from recent trials and highlight both the therapeutic potential and adherence challenges associated with these dietary approaches. They conclude that nutritional therapy, while less established in adults than in pediatrics, represents an increasingly important adjunct to conventional treatments.
The topic is timely and clinically relevant, as dietary management in adult Crohn’s disease remains underexplored compared to pediatric cohorts. The manuscript is overall well referenced and structured, with clear sections devoted to different nutritional strategies. That said, the tone of the review is more descriptive than analytical, and the critique of available evidence could be sharper.
Specific strengths: Comprehensive coverage of the main nutritional interventions currently studied in Crohn’s disease; integration of mechanistic insights (e.g., microbiome, barrier integrity) with clinical outcomes; attention to cultural and adherence factors, which are often neglected in reviews.
Major issues: The review frequently summarizes studies without adequately dissecting their methodological limitations (small sample sizes, lack of endoscopic endpoints, heterogeneity in populations). A more incisive appraisal would improve the manuscript’s value; while several diets are described in parallel, there is little effort to compare their relative strengths, limitations, and real-world applicability. A more direct head-to-head discussion would help clinicians; the review mentions IBD-MAID and the role of additives, but this could be expanded with a dedicated discussion on low emulsifier diets. There is growing preclinical and translational evidence that emulsifiers such as polysorbate 80 and carboxymethylcellulose may exacerbate gut inflammation and dysbiosis. Given the clinical momentum in this area, this omission feels important; the review cites STRIDE II, but the practical implications for integrating diet into treat-to-target strategies could be more explicitly stated.
We sincerely thank the reviewer for the positive feedback on our work and for the insightful suggestions. In the revised version, we have adopted a more critical approach throughout each section, highlighting the main methodological limitations of the available studies (such as small sample sizes, lack of endoscopic endpoints, and heterogeneity of study populations) as well as the practical challenges of applying different diets in real-life settings. We have also aimed to provide more direct comparisons between dietary approaches, emphasizing their respective strengths and weaknesses in a clinically relevant manner. Furthermore, the revised manuscript now includes a dedicated discussion on emulsifiers, with reference to recent preclinical and translational evidence regarding additives such as polysorbate 80 and carboxymethylcellulose. Finally, regarding STRIDE-II, we have expanded the discussion to clarify our view that dietary interventions may represent a valuable adjunct primarily for the short-term clinical targets outlined in STRIDE-II, but also for intermediate and long-term targets, including improvements in quality of life.
Minor issues: The abstract and introduction could be tightened to reduce repetition; some sections, particularly on the Specific Carbohydrate Diet, are overly detailed given the limited evidence in adults, while other sections (Mediterranean diet, emulsifiers) are comparatively thin; table 1 is useful but could benefit from clearer definitions of CR/BR/ER for non-specialist readers.
In the revised manuscript, we have streamlined both the abstract and the introduction to reduce repetition and improve readability. The section on the Specific Carbohydrate Diet has been shortened, with fewer details included, while we have expanded the sections on the Mediterranean diet and on emulsifiers to provide a more balanced and comprehensive discussion. Finally, Table 1 has been revised with clear definitions of CR (clinical response), BR (biochemical remission), and ER (endoscopic remission) and specifying the definitions of these outcomes vary between studies, with high heterogeneity.
Reviewer 3 Report
Comments and Suggestions for Authors
The manuscript is a review that addresses the topic of nutritional therapy in the treatment of adults with Crohn’s disease (CD). The authors focused on six types of specific diets in patients with CD, effectively communicating the pros and cons of each. The manuscript is well written in terms of the English language. The topic is interesting for researchers and clinicians, providing sufficient background to understand its message. However, it requires improvement by reviewing a few issues:
- Although it does not go into depth on each type of diet, the review is clear and well structured, with a single integrative table. However, the authors do not have any figure showing the pathophysiological mechanisms by which each diet improves the clinical, biochemical, or endoscopic response in patients with CD.
- The inclusion/exclusion strategy of the articles considered for this review and the search terms used are not defined. The authors mention in passing in the abstract that “We conducted a comprehensive PubMed literature search up to May 2025…”. A clear PRISMA flow diagram is required.
- Authors should clearly state what they are referring to when discussing biochemical response (BR). Are they referring to serum CRP and/or fecal calprotectin levels, or also other inflammatory markers?
- Try to be consistent with the abbreviations throughout the text (e.g., in the penultimate paragraph of chapter 5, CD and IBD have already been abbreviated previously). Also, in the table, RCT is not defined as an abbreviation.
- In the table, in the columns with therapeutic responses, a check mark or an X ​​appears. When X appears, authors must distinguish between no data/no response.
Overall, even though it is a brief review, I enjoyed reading this well-organized manuscript.
Author Response
Reviewer 3: The manuscript is a review that addresses the topic of nutritional therapy in the treatment of adults with Crohn’s disease (CD). The authors focused on six types of specific diets in patients with CD, effectively communicating the pros and cons of each. The manuscript is well written in terms of the English language. The topic is interesting for researchers and clinicians, providing sufficient background to understand its message. However, it requires improvement by reviewing a few issues
We thank the reviewer for the positive feedback
1.Although it does not go into depth on each type of diet, the review is clear and well structured, with a single integrative table. However, the authors do not have any figure showing the pathophysiological mechanisms by which each diet improves the clinical, biochemical, or endoscopic response in patients with CD.
We thank you for this helpful suggestion. As requested, we have added Figure 1 to illustrate the pathophysiological mechanisms through which different diets may contribute to clinical outcomes in patients with Crohn’s disease.
2.The inclusion/exclusion strategy of the articles considered for this review and the search terms used are not defined. The authors mention in passing in the abstract that “We conducted a comprehensive PubMed literature search up to May 2025…”. A clear PRISMA flow diagram is required.
We thank the Reviewer for this important observation. We would like to clarify that our manuscript is a narrative review, not a systematic review. Therefore, the use of a PRISMA flow diagram was not performed. Nevertheless, we fully agree on the importance of transparency in the selection process. In response to the Reviewer’s comment, we have now: 1. Clearly stated in the introduction that this is a narrative review; 2. Provided the keywords used for the PubMed literature search; 3. Added a brief description of the inclusion and exclusion criteria applied in selecting the studies.
3.Authors should clearly state what they are referring to when discussing biochemical response (BR). Are they referring to serum CRP and/or fecal calprotectin levels, or also other inflammatory markers?
In the manuscript, when referring to biochemical response (BR), we are specifically considering the most used and traditional biomarkers in IBD, namely serum C-reactive protein and fecal calprotectin. We have now made this explicit in the text.
4.Try to be consistent with the abbreviations throughout the text (e.g., in the penultimate paragraph of chapter 5, CD and IBD have already been abbreviated previously). Also, in the table, RCT is not defined as an abbreviation.
We acknowledge the inconsistencies with abbreviations and thank you for pointing them out. We have carefully revised the manuscript to ensure consistency throughout, including the suggested corrections in chapter 5.
5.In the table, in the columns with therapeutic responses, a check mark or an X ​​appears. When X appears, authors must distinguish between no data/no response.
In the table 1 of the revised version, we now clearly distinguish between “no data available” and “no response,” to avoid any possible ambiguity for readers.
Round 2
Reviewer 1 Report
Comments and Suggestions for Authors
The authors addressed all the issues.